# Alkaline-based aqueous sodium-ion batteries for large-scale energy storage

Han Wu[1,3], Junnan Hao[1,3], Yunling Jiang[1], Yiran Jiao[1], Jiahao Liu[1], Xin Xu[1], Kenneth Davey [1], Chunsheng Wang [2] & Shi-Zhang Qiao [1] ✉

Aqueous sodium-ion batteries are practically promising for large-scale energy storage, however energy density and lifespan are limited by water decomposition. Current methods to boost water stability include, expensive fluorine-containing salts to create a solid electrolyte interface and addition of potentially-flammable co-solvents to the electrolyte to reduce water activity. However, these methods significantly increase costs and safety risks. Shifting electrolytes from near neutrality to alkalinity can suppress hydrogen evolution while also initiating oxygen evolution and cathode dissolution. Here, we present an alkaline-type aqueous sodium-ion batteries with Mn-based Prussian blue analogue cathode that exhibits a lifespan of 13,000 cycles at 10 C and high energy density of 88.9 Wh kg$^{-1}$ at 0.5 C. This is achieved by building a nickel/carbon layer to induce a $H_3O^+$-rich local environment near the cathode surface, thereby suppressing oxygen evolution. Concurrently Ni atoms are in-situ embedded into the cathode to boost the durability of batteries.

The growing demand for large-scale energy storage has boosted the development of batteries that prioritize safety, low environmental impact and cost-effectiveness[1–3]. Because of abundant sodium resources and compatibility with commercial industrial systems[4], aqueous sodium-ion batteries (ASIBs) are practically promising for affordable, sustainable and safe large-scale energy storage. However, energy density and cycling stability are limited because of the narrow electrochemical stability window of 1.23 V for water. Additionally, the accumulation of flammable hydrogen (H$_2$) from water decomposition during cycling compromises battery safety and restricts the development of ASIBs. A common method for improving the performance of aqueous batteries is to use expensive fluorine-containing salts to create a solid-electrolyte interphase (SEI)[5] that suppresses the hydrogen evolution reaction (HER) and increases the electrochemical window of the electrolyte. However, the high solubility of SEI components including LiF, NaF and Na$_2$CO$_3$ limits durability[6]. In addition the high cost of fluorine-containing salts significantly compromises cost-effectiveness of aqueous batteries. Another method involves using co-solvents[7], including polymers, to improve water stability of the electrolyte. A drawback however is these significantly increase the viscosity of the electrolyte, making it practically difficult to match with

high-loading electrodes for commercial application. The potential flammability of organic co-solvents can increase safety risk of aqueous batteries. Alternative methods to boost water stability of aqueous batteries whilst maintaining cost-effectiveness and safety is therefore of practical interest.

Compared with conventional aqueous neutral electrolytes, alkaline electrolytes thermodynamically suppress HER on the anode based on the Pourbaix diagram for water[8]. Whereas shifting the electrolyte from near neutrality to alkalinity intensifies the oxygen evolution reaction (OER) on the cathode[9]. The high concentration of OH$^-$ in electrolytes limits selection of cathodes because of the interaction of transition metal-based electrodes with OH$^-$, leading to the deterioration of electrode structures, especially for Mn-based Prussian blue analogues (PBAs) cathodes[10]. As widely used cathode materials, PBAs have been reported in traditional aqueous batteries with advantages of non-toxicity, low cost and high energy density[5–7]. However application in alkaline electrolytes is restricted because of strong Jahn-Teller effects induced by the redox couples of Mn$^{2+}$/Mn$^{3+}$ together with the dissolution of Fe which dissolve in the alkaline electrolyte as an Fe(CN)$_6^{3/4-}$ complex[10,11]. In consequence, PBA-based alkaline ASIBs are not developed yet.

[1]School of Chemical Engineering, The University of Adelaide, Adelaide, SA 5005, Australia. [2]Department of Chemical and Biomolecular Engineering, University of Maryland, College Park, MD 20742, USA. [3]These authors contributed equally: Han Wu, Junnan Hao. ✉e-mail: s.qiao@adelaide.edu.au

Here we report a hydrogen-free alkaline ASIB based on a Mn-based PBA cathode ($Na_2MnFe(CN)_6$, NMF), $NaTi_2(PO_4)_3$ (NTP) anode, and an affordable alkaline electrolyte of fluorine-free sodium perchlorate ($NaClO_4$) where cost is significantly less than commonly-used sodium triflate and sodium bis(tri-fluoromethylsulfonyl)imide in highly concentrated electrolytes. As illustrated in Fig. 1a, the alkaline electrolyte suppresses HER at the anode. Via coating a commercially available nickel/carbon (Ni/C) nanoparticle-based layer on the NMF cathode, a $H_3O^+$-rich local environment formed near the cathode surface. This $H_3O^+$-rich local environment resulted from the irreversible formation of $Ni(OH)_2$ and reversible $Ni(OH)_2$/NiOOH redox (confirmed by in-situ Attenuated Total Reflectance Infrared (ATR-IR), and *operando* synchrotron X-ray powder diffraction, XRPD) that significantly reduces OER and electrode dissolution. Additionally, partial Ni atoms in the coating are in-situ embedded in the cathode to stabilize the NMF structure in alkaline media, as confirmed via *operando* Raman and high-angle annular dark-field scanning transmission electron microscopy (HAADF-STEM).

## Results

### Electrochemical performance of alkaline NMF//NTP coin cells

Prepared NMF, NTP and commercial Ni/C powders were subjected to X-ray diffraction (XRD, Supplementary Figs. 1–3), evidencing good crystallinity for applications in batteries. The impact of salt concentrations in electrolytes on HER was established via in-situ differential electrochemical mass spectrometry (DEMS) in $NaClO_4$ electrolytes with selected salt concentrations. Findings confirm that, without forming a reliable SEI, increasing electrolyte concentration does not change the onset potential for HER (Supplementary Fig. 4).

Importantly, increasing the alkalinity of electrolyte suppresses HER (Supplementary Fig. 5a-c) that contributes to lower over-discharge (caused by HER) of NTP anode (Supplementary Fig. 6a−c). However, the increased alkalinity of electrolyte drives OER (Supplementary Fig. 5d) and increases dissolution rate of Fe and Mn elements[12] of the NMF cathode, leading to poor cycling stability (Supplementary Fig. 6d, e). In comparison, following coating of the Ni/C layer on the NMF electrode (thickness: *ca.* 1 μm, Supplementary Fig. 7), cycling stability is significantly boosted (Supplementary Fig. 6f), as confirmed

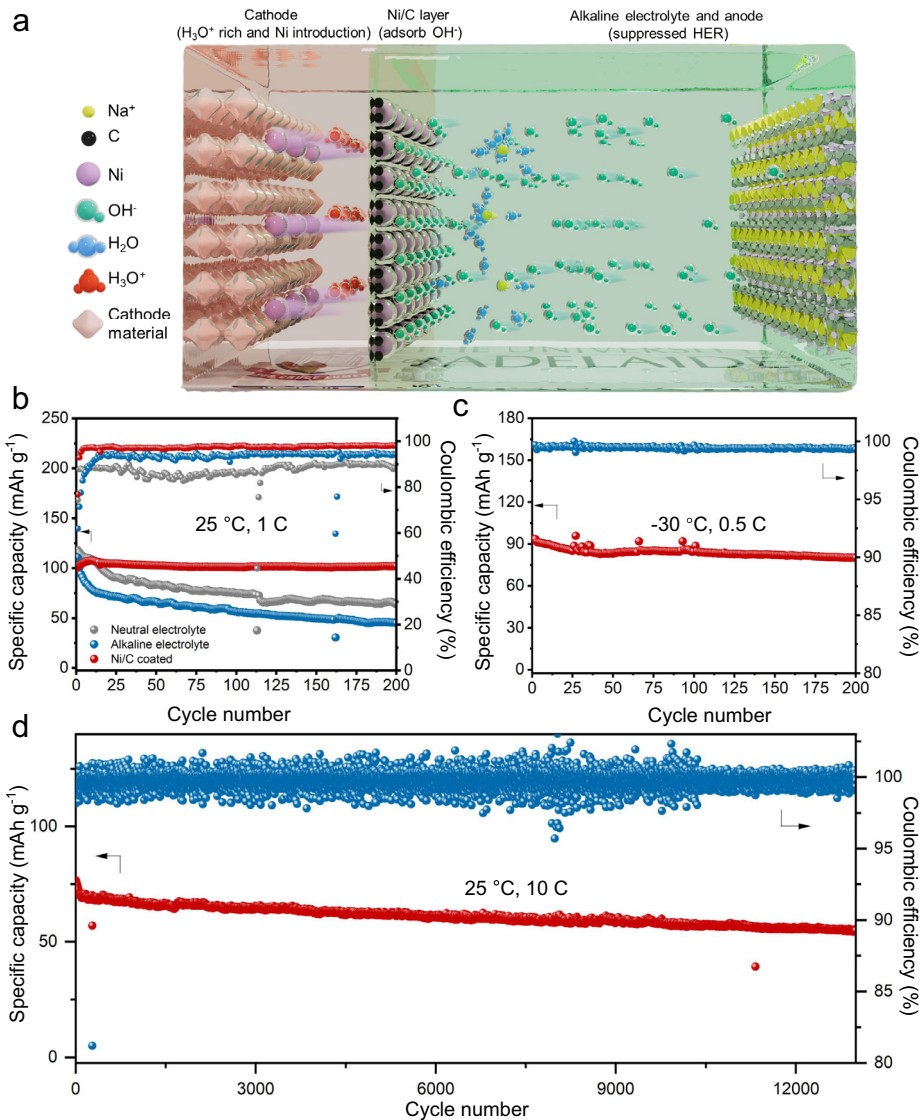

**Fig. 1 | Electrochemical performance of NMF//NTP coin cells in the range of 0.5 to 2.2 V. a** Design concept for the alkaline aqueous battery. **b** Cycling performance for three cells at 1 C. **c** Cycling performance for NMF//NTP cell with Ni/C coating at 0.5 C at a low temperature of −30 °C. **d** Long-term cycling stability for NMF//NTP full cell with Ni/C coating at 10 C, 1 *C* = 118 mA g⁻¹, based on NMF.

via unchanged electrolyte colour (Supplementary Fig. 6g–i) together with the suppressed Fe dissolving concentration in the electrolyte evidenced via inductively coupled plasma mass spectrometry (ICP-MS) (Supplementary Fig. 8).

The performance of NMF//NTP full cells using a neutral electrolyte or a alkaline electrolyte with/without Ni/C coating were tested in a wide charging voltage range of 0.5 to 2.2 V. The NMF//NTP full cell with Ni/C coating exhibits a highly boosted rate performance and higher average discharge voltage than those without Ni/C coating, contributing to the fast-charge ability and high average voltage of the battery (Supplementary Figs. 9a, b). Figure 1b compares the cycling performances of NMF//NTP full cells under three different conditions at 1 C. Batteries without Ni/C coating in both neutral and alkaline electrolytes exhibits a rapid capacity decay with capacity retention of <60% following 200 cycles, whereas the alkaline-based battery with Ni/C coating exhibits a greater retention of *ca.* 100%. Significantly, the electrolyte exhibited a lower freezing point in comparison with reported highly concentrated electrolytes[5,11], allowing the battery to function at low temperatures (Supplementary Fig. 10). The battery with Ni/C coating therefore exhibits a capacity retention of 91.3% after 200 cycles at 0.5 C under −30 °C (Fig. 1c). Importantly, this full cell exhibits a record lifespan of

>13,000 cycles with a high capacity retention of 74.3% at 10 C (Fig. 1d) in alkaline electrolyte, surpassing reported performances of many aqueous batteries[13].

## Pouch cell performance and comparison

To simulate commercial requirements for large-scale energy storage, a Ni/C coated NMF//alkaline electrolyte//NTP pouch cell was assembled with an electrode loading of *ca.* 20 mg cm⁻². This pouch cell exhibits a high capacity retention of 85% following 1,000 cycles at 500 mA g⁻¹ (Fig. 2a). Additionally, the mass loading of the single electrode can be increased to >30 mg cm⁻² because of the low viscosity of the alkaline electrolyte of 6.0 mPa·s (Supplementary Table 1). With this high loading, NMF//NTP pouch cell exhibits stable cycling life with a capacity retention of *ca.* 100% within 200 cycles at 300 mA g⁻¹ (Fig. 2b). This large-size pouch cell additionally shows high stability under 'harsh' conditions of cutting and immersion in water (Supplementary Movies 1–2, Supplementary Fig. 11a–c and Fig. 2c). Significantly, the cut pouch cell powers continuously a digital hygrometer thermometer in water for >20 h (Supplementary Movie 3, Supplementary Fig. 11d–f and Fig. 2d). This finding confirms that the battery is resistant to electrolyte leakage and can withstand significant damage in the high-humidity

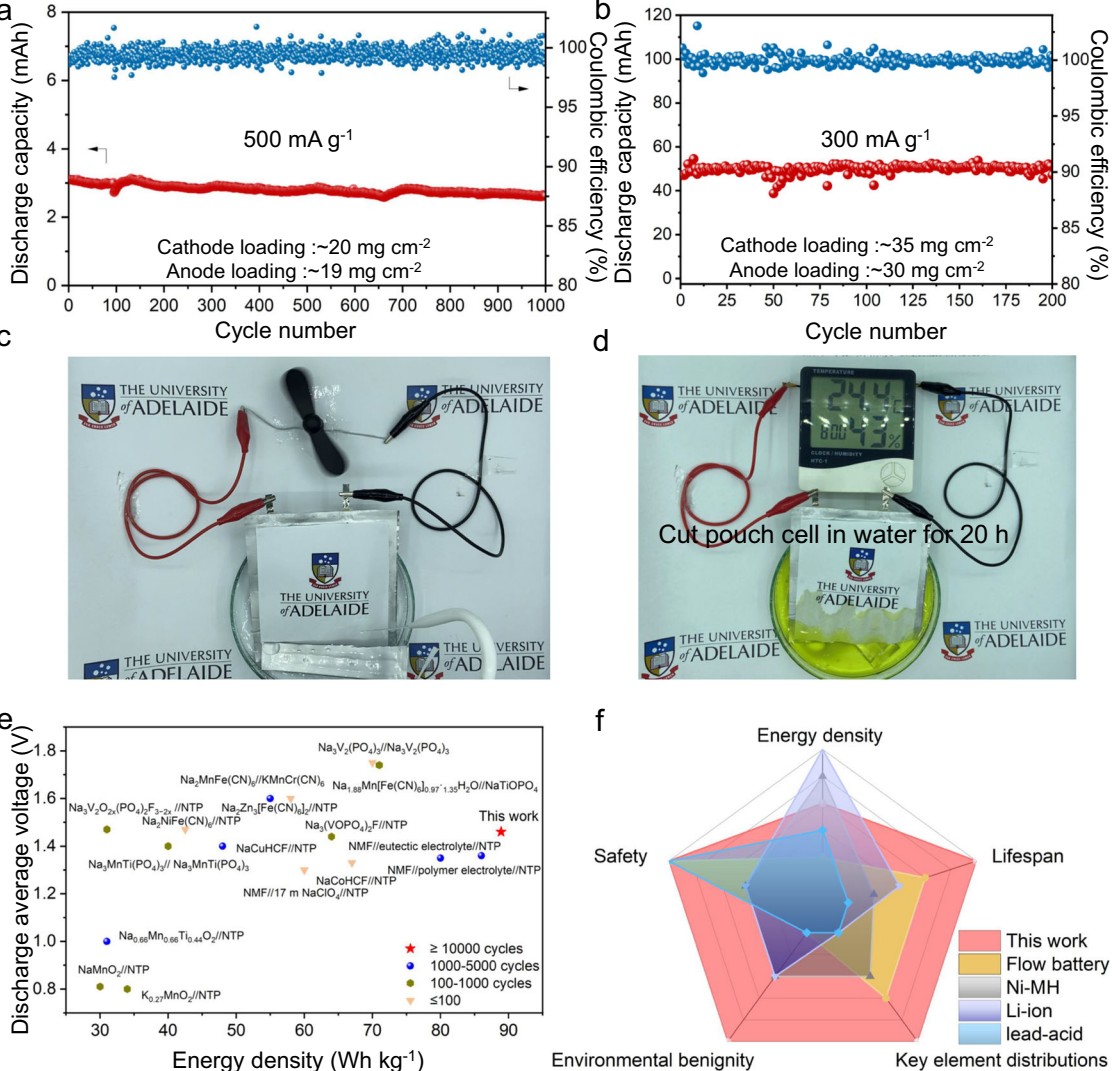

**Fig. 2 | Electrochemical performance for NMF//NTP pouch cells and comparison with selected reports. a** Cycling performance for NMF//NTP pouch cell at a current density of 500 mA g⁻¹. **b** Cycling performance for NMF//NTP pouch cell at 300 mA g⁻¹. **c** Digital photograph of a cut pouch cell to power a fan in water.

**d** Digital photograph of a cut pouch cell to power a humidity clock in water.
**e** Comparison of lifespan and energy density with reported ASIBs. **f** Comparison of present work with commercial batteries (as quantified in Supplementary Table 4).

environment. The cycled pouch cell exhibits no apparent volume changes, evidencing that there is no significant gas evolution during cycling (Supplementary Fig. 12). The battery therefore exhibited high safety (low risk) for practical application in energy storage and underwater electrical equipment. Importantly, the batteries exhibited a high cycling stability and discharge capacity under a low positive/negative capacity ratio of 1.06. Under such a low capacity ratio, and 0.5 C charging rate, a high energy density amongst ASIBs of 88.9 Wh kg$^{-1}$ is achieved (Fig. 2e and Supplementary Table 2). The cell-level energy density of the new ASIB based on a virtual cell configuration with realistic parameters was computed, Supplementary Table 3. The compacted density and porosity of the electrode, the amount of electrolyte, size of the current collector, tab and package, were determined using empirical parameters derived from the reported literature assuming a 10-layer pouch cell geometry. The energy density was computed via dividing the total energy by the total mass of the pouch cell. The predicted battery energy density is *ca.* 61 Wh kg$^{-1}$. Compared with reported electrochemical storage devices, this new battery exhibits significant advantages including, use of abundant elements (such as Fe, Mn and Ti), high safety (high tolerance of high

humidity environment), environmental benignity (non-poisonous electrolyte) and a long lifespan (Fig. 2f and Supplementary Table 4).

## Origin of $H_3O^+$-rich local environment

To determine the underlying factors for high performance of the alkaline ASIB, the interface structure with Ni/C coating was assessed via in-situ ATR-IR spectroscopy. The carbon coating was taken as a control group to eliminate any influences of carbon and Nafion-Na support. For the electrode modified by pure carbon, the spectra exhibited no apparent change despite being charged to 1.3 V (*vs.* Ag/AgCl), evidencing that the carbon and support do not change local environment of the cathode surface (Fig. 3a, the function of carbon in Ni/C coating is discussed in Supplementary Text S1 and Supplementary Fig. 13). In contrast, with Ni/C modification, new peaks at 1798 and 2032 cm$^{-1}$ appear when the potential is >0.6 V, attributed to two asymmetric O-H stretching modes of $H_3O^+$ ($\nu_{H_3O^+}^{a1}$ and $\nu_{H_3O^+}^{a2}$). Peaks for the resonance state for the asymmetric O-H stretching modes in $H_3O^+$ ($\nu_{H_3O^+}^{a2,r+u}$) at 2223 cm$^{-1}$ and the umbrella vibration for $H_3O^+$ ($\nu_{H_3O^+}^{u}$) at 1230 cm$^{-1}$ are also visible[14,15] confirming that the $H_3O^+$ accumulation on the electrode

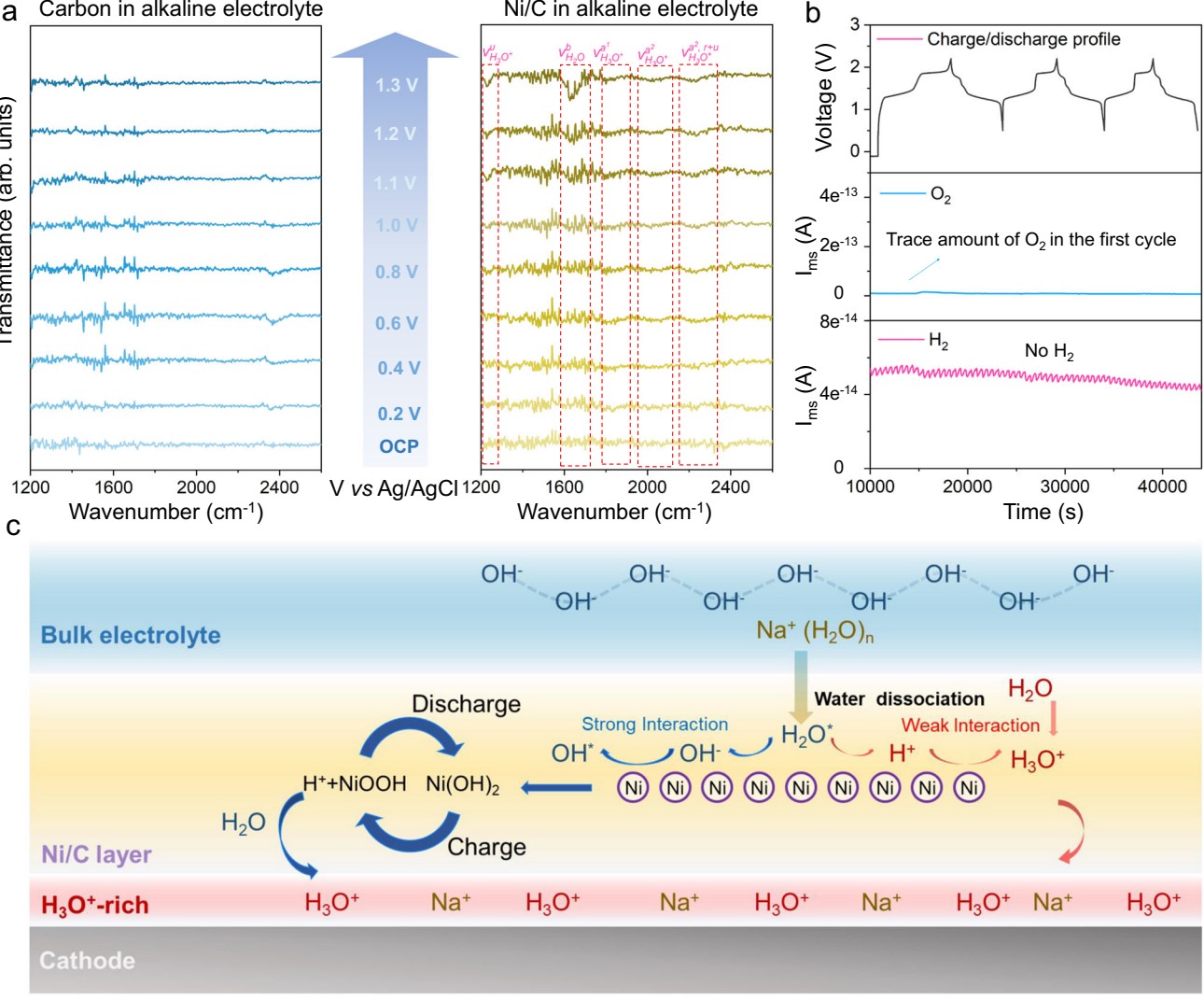

**Fig. 3 | Generation of $H_3O^+$-rich micro-environment. a** ATR-FTIR spectra for pure carbon and Ni/C at different potentials. Fundamental excitations of $H_3O^+$, denoted $\nu_{H_3O^+}^{u}$ (umbrella vibration), $\nu_{H_3O^+}^{a1}$ (asymmetric O-H stretching), and resonance states between fundamental excitation of asymmetric O-H stretching and combination tones (r + u) denoted $\nu_{H_3O^+}^{a2,r+u}$, where r and u represent, respectively,

frustrated rotation and umbrella vibration. Bending of $H_2O$ is denoted $\nu_{H_2O}^{b}$. **b** *Operando* DEMS findings to determine $H_2$ and $O_2$ evolution during NMF//NTP battery cycling at 0.5 to 2.2 V at 0.5 C. **c** Schematic for $H_3O^+$ accumulation mechanism on electrode surface coated with Ni/C in the alkaline electrolyte.

surface is induced by Ni nanoparticles. *Operando* DEMS was used to analyze water decomposition in this alkaline battery during cycling. The battery without Ni/C coating exhibits HER and OER concurrently at a low positive/negative capacity ratio in the neutral electrolyte (Supplementary Fig. 14). However, following coating Ni/C on the NMF cathode in alkaline electrolyte, both HER and OER are not apparent, except for trace $O_2$ at the first cycle before activating the surface coating (Fig. 3b). It is concluded therefore that the $H_3O^+$-rich local environment induced by Ni/C protective layer suppresses OER in the alkaline electrolyte, whilst the alkaline electrolyte retards HER.

The $H^+$ accumulation mechanism on the electrode surface with Ni/C is illustrated in Fig. 3c. $OH^-$ ions exhibit greater adsorption energy on the Ni surface compared with $H^+$ with an applied positive voltage, confirmed via density functional theory (DFT) simulation (Supplementary Table 5). This leads to a localized decrease in $OH^-$ concentration under the protective layer during charging. Together with the increased voltage, Ni gradually undergoes oxidation, transforming into $Ni(OH)_2$ (Supplementary Fig. 15a, b). This further consumes local $OH^-$, causing a subsequent decrease in pH. The other component in the coating layer, the Nafion-based polymer that is known as the cationic membrane, inhibits the diffusion of $OH^-$ from the bulk electrolyte to the electrode surface (Supplementary discussion Text S2 and Supplementary Figs. 16–17). Therefore, the reduced pH environment is maintained on the cathode surface. $Ni(OH)_2$ subsequently converts to NiOOH (Supplementary Fig. 15b–d) which generates additional $H^+$ ions. These $H^+$ ions generated during the reaction remain un-neutralized because of the reduced local pH, leading to a $H_3O^+$-rich local environment. The $Ni(OH)_2$/NiOOH transformation is highly reversible (the capacity provided by Ni/C was computed, Supplementary Fig. 18), as confirmed via XRPD and soft X-ray absorption spectra (Supplementary Fig. 19 and Supplementary discussion Text S3), resulting in a dynamic equilibrium boost the stability of $H_3O^+$-rich local environment under high voltages.

## Ni-substitution

In addition to increased OER, the improved alkalinity of electrolyte compromises the cycling stability of PBA-based cathode material (without Ni/C coating). Therefore, in addition to inducing $H_3O^+$-rich environment, the Ni/C coating also results in the substitution of Ni to fill Mn vacancies, thereby stabilizing the PBA cathode. The working principle for this system is illustrated in Fig. 4a. During charging, the surface of the Mn-based PBA cathode exhibits Mn dissolution, leading to the generation of Mn vacancies[16]. In the unprotected system, the continuous dissolution of Mn ions leads to structural collapse and adverse effects on the cyclic stability of the battery (Fig. 4b, c). However, in the Ni/C protected system, the in-situ substituted Ni atoms balance tiny structural disturbances caused by Mn dissolution as evidenced by the stable discharge plateau of NMF//NTP battery following the coating[10,17,18] (Fig. 4d). The oxidation of Ni during discharge produces $Ni^{2+}$ ions. These $Ni^{2+}$ ions gradually enter the crystalline framework to fill the Mn vacancies by forming Ni-N bonds on cycling. DFT computations (Supplementary Fig. 20) for Ni substitution were carried out to determine a $\Delta E$ value of $-8.06$ eV, evidencing that Ni substitution is spontaneous.

Ni substitution was further confirmed via *operando* Raman spectra for Ni/C coated NMF cathodes cycled in alkaline electrolyte (Fig. 4e). Prior to cycling, two peaks at 2089 and 2124 $cm^{-1}$ are apparent, corresponding to $Fe^{2+}-CN-Mn^{2+}$ and $Fe^{2+}-CN-Mn^{3+}$ vibrations, respectively[19]. Following charging to 1.89 V both peaks disappear, evidencing the transformation of $Fe^{2+}$ to $Fe^{3+}$ and $Mn^{2+}$ to $Mn^{3+}$. Importantly, following charging to 2.2 V, a new weak peak appears at 2195 $cm^{-1}$ corresponding to $Fe^{3+}-CN-Ni^{2+}$ [20]. This finding confirms that the introduction of Ni atoms in NMF particles follows the transformation of $Mn^{2+}$ to $Mn^{3+}$. Following discharging to 0.5 V, peaks for

$Fe^{2+}-CN-Mn^{2+}$ and $Fe^{2+}-CN-Mn^{3+}$ shift to 2092 and 2128 $cm^{-1}$ and a new peak appears at 2164 $cm^{-1}$ that is assigned to $Fe^{2+}-CN-Ni^{2+}$ [20], confirming the introduction of Ni.

The introduction of Ni was also confirmed via STEM-EDS mapping (Fig. 4f). There is a new peak belonging to Ni element in the spectrum. Additionally, the EDS line scan spectra for a single NMF particle confirm that Ni atoms are introduced into the edge of particles to suppress the dissolution of inner Mn atoms (Fig. 4g). The STEM-energy-dispersive spectroscopy (STEM-EDS) mappings for NMF cathode with Ni/C coating following 1st, 5th and 20th cycles (Supplementary Table 6 and Supplementary Fig. 21) evidence that the content of Ni in NMF particles is stable after the first cycle, confirming that the introduction of Ni into NMF cathode reaches an equilibrium in the first cycle to give long-term stability to the battery.

To demonstrate the cathode structure stability following Ni/C protection, other characterizations were conducted. Digital photographs of the PBA electrodes following cycling shows that the unprotected electrodes exhibit significant metal ion dissolution in both neutral and alkaline conditions (Supplementary Fig. 22). In comparison, the dissolution phenomenon is significantly mitigated when the Ni/C protective coating is applied. TEM analyses evidence that the structure of the PBA cathode undergoes significant structural damage following cycling in the alkaline condition (Supplementary Fig. 23). STEM-EDS confirms that the electrode exhibits Mn dissolution in the neutral media, whilst the concurrent dissolution of both Fe and Mn occurs in alkaline electrolyte (Supplementary Fig. 24). However, following application of the protective coating, the electrode dissolution is significantly suppressed resulting from in-situ Ni substitution.

The boosted structural stability of NMF cathode following introduction of inert Ni atoms in alkaline electrolyte was confirmed via XRPD patterns during charge/discharge. The structural evolutions of NMF in the first cycle is shown in Fig. 5a and Fig. 5b presents the 2D contour map for NMF reflection. The highly reversible structure evolution during charge and discharge can be easily observed in these figures. Additionally, the Rietveld refinements of NMF with/without Ni/C coating after 1st cycle evidence that both electrodes exhibit cubic phases with Fm-3m space group and $a = b = c$ (Fig. 5c, d and Supplementary Table 7). The lattice parameters for Ni/C coated NMF (5.28161 Å) are greater than that for uncoated NMF (5.26358 Å). This finding is attributed to the introduction of Ni in NMF during cycling. An increase in a (b, c) contributes to a boosted rate performance for the cathode, which, significantly, is consistent with our findings. Importantly, compared with the deteriorated structure of uncoated NMF after 1st and 3rd cycles (Fig. 5e), the overlapping patterns for Ni/C coated NMF following 1st and 3rd cycles confirm that the excellent stability of NMF and Ni introduction occurs at 1st cycle, because otherwise, continuous Ni introduction will change the XRPD pattern (Fig. 5f).

To assess the possible universality of the new electrode modification method in alkaline batteries, the Co/C nanoparticle was used to build the cathode coating. Similar with Ni nanoparticles, Co can be oxidized to $Co(OH)_2$ in alkaline media and, it exhibits a reversible redox pair of $Co(OH)_2$/CoOOH, together with the ability to in-situ substitute the Mn atom. As a result, good stability of the battery with Co/C coating is achieved (Supplementary Fig. 25). This finding provides evidence for the universality of creating $H_3O^+$-rich cathode surfaces and in-situ optimizing the NMF structure by building metal nanoparticle coating to boost the performance of Mn-based PBA cathode in an alkaline environment.

## Discussion

A new aqueous battery system that is different to traditional ASIBs based on near neutral electrolyte, is presented with a fluorine-free alkaline electrolyte to suppress $H_2$ evolution on the anode and a Ni/C coating to obviate $O_2$ evolution and electrode dissolution on the cathode. This system achieves long cycling stability of 13,000 cycles

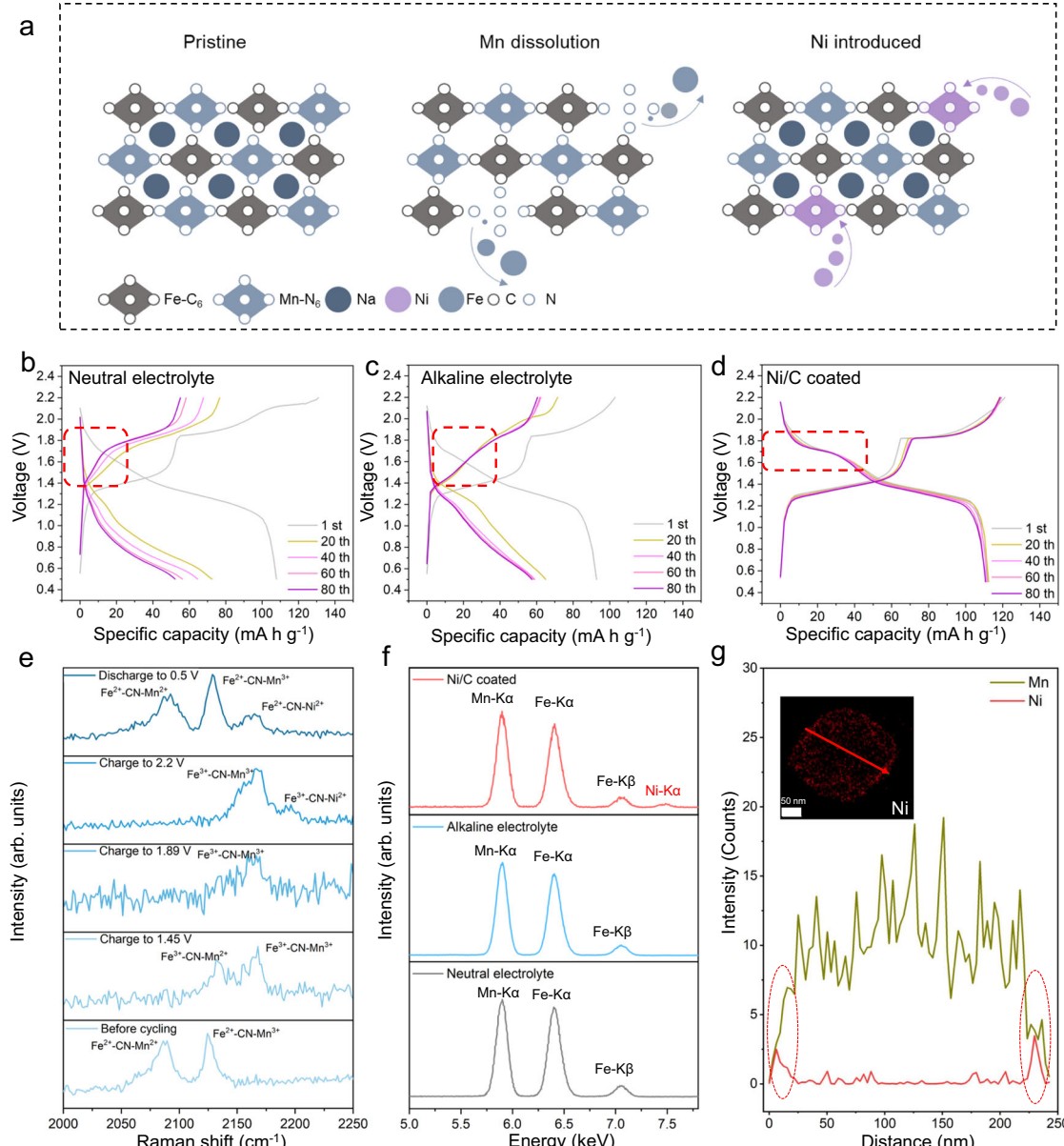

**Fig. 4 | Determination of reaction mechanism and in-situ Ni substitution.**
**a** Schematic for Ni/C coating for mitigating structural instability because of Mn dissolution in NMF cathode. Charge/discharge curves for NMF//NTP cells in (**b**) Neutral electrolyte (**c**) Alkaline electrolyte and (**d**) Alkaline electrolyte with Ni/C coating. **e** *Operando* Raman spectra for Ni/C coated NMF cathode cycled in alkaline electrolyte. **f** STEM-EDS spectra taken from NMF electrodes following cycling in neutral, alkaline electrolyte and alkaline electrolyte with Ni/C coating. **g** STEM line scan for cycled NMF cathodes with Ni/C coating. Inset shows Ni mapping for NMF.

and high energy density of 88.9 Wh kg$^{-1}$ in the alkaline electrolyte through the Ni/C coating-induced $H_3O^+$-rich local environment and in-situ electrode Ni modification. A pouch cell, assembled with a high electrode loading of >30 mg cm$^{-2}$ can maintain a capacity retention of *ca.* 100% following 200 cycles, confirming excellent safety despite being cut and immersed in water. This aqueous alkaline battery design appears universal by extending to Co/C and exhibits practical prospects for high energy density via coupling with other lower redox potential anodes (Supplementary Fig. 26). Importantly, this method can be expanded flexibly to selected aqueous batteries to boost practical applications.

## Methods
### Materials
The $Na_2MnFe(CN)_6$ (NMF) cathode and $NaTi_2(PO_4)_3$ (NTP)/C anode were synthesized based on reported methods[21]. To be specific, $Na_4Fe(CN)_6$ (5 mmol; Sigma–Aldrich, 99 %) and NaCl (15 g;

Sigma–Aldrich, 99 %) were dissolved in 100 mL de-ionized water to form solution A. Meanwhile, $MnCl_2$ (5 mmol; Sigma–Aldrich, 99 %) was dissolved in 50 mL deionized water to create solution B. Solution B was slowly added into solution A over 20 min with stirring. The resulting suspension was stirred for an additional 2 h to complete the reaction. After resting for 12 h, the solid phase was separated via centrifugation, washed three times with 30 mL water, dried under vacuum, and finely ground into a powder. This powder was stored in a vacuum oven at 110 °C for 24 h before utilization. Additionally, NTP/C was synthesized through the following method. Initially, 2.5 mmol sodium acetate trihydrate and 7.5 mmol ammonium dihydrogen phosphate were dissolved in 100 mL deionized water to form solution C. Subsequently, 0.4 g pyrrolidone (PVP, Sigma Aldrich) and 5 mmol titanium(IV) butoxide (Sigma–Aldrich ≥ 98 %) were dissolved in 25 mL anhydrous ethanol (Sigma–Aldrich ≥ 99.9 %) to create solution D. Solution D was added to solution C under stirring for 3 h. The precursor was obtained

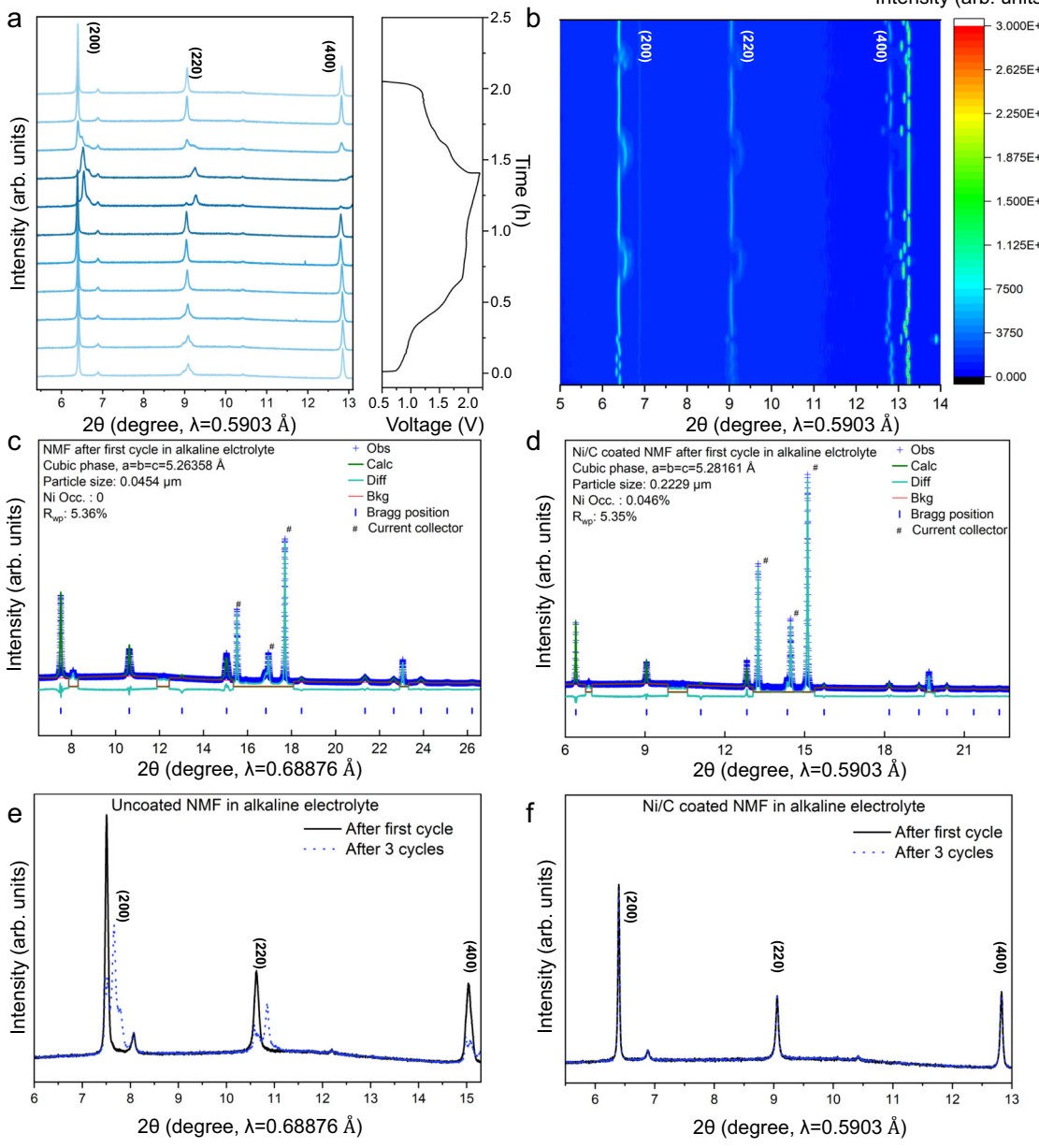

**Fig. 5 | *Operando* structure analyses for NMF cathode during battery cycling at 0.5 to 2.2 V. a** (200), (220) and (400) reflections of synchrotron *operando* XRPD pattern. **b** 2D contour plot for *operando* XRPD for the first 3 cycles. Structural and Rietveld refinements for NMF cathode following 1st cycle in (**c**) Alkaline electrolyte and (**d**) Alkaline electrolyte with Ni/C coating. Comparison of XRPD pattern following 1st cycle and 3rd cycle in (**e**) alkaline electrolyte and (**f**) alkaline electrolyte with Ni/C coating.

after evaporation to remove solvents. Finally, the NTP/C composite was achieved by calcining the precursor at 800 °C for 12 h under an argon atmosphere, with a temperature rise rate of 5 °C min⁻¹. The carbon content of the NTP/C materials was ~5 wt %. The Nafion-Na was prepared through the neutralization reaction of 10 mL Nafion (purchased from *DuPont*, D520, 5 wt%) with 0.01 M NaOH solution drop-by-drop. Product was collected following the removal of the solvent at 60 °C. 20% Ni/C and Co/C were purchased from *Fuel Cell Store*.

### Electrode preparation

The NMF electrode was prepared via mechanically mixing 80 m/m% NMF, 10 m/m% Super-P and 10 m/m% polytetrafluoroethylene (PTFE) binder dispersed in ethanol solvent (AR). The mixture was pressed on a Ti-mesh at a pressure of 6 MPa and dried at 70 °C for 2 h in a vacuum oven. The NTP electrode was prepared *via* the same procedure as the NMF electrode by using 80 m/m% NTP/C (*ca.* 5% C), 10 m/m% Super-P and 10 m/m% PTFE. The mass loading for electrodes was *ca.* 10 mg cm⁻². The mass ratio of the anode and cathode in the coin cell was *ca.* 1/1.06. The electrodes in the pouch cell were prepared with the same method. The mass ratio of the anode and cathode in the pouch cell was *ca.* 1/1.15. The specific capacity computation was based on the mass of cathode. The energy density was computed from:

$$\text{Energy density} = \frac{\text{Discharge energy}}{\text{Total mass of anode and cathode}} \quad (1)$$

### Electrolyte preparation

A neutral electrolyte with a concentration of 17 $m$ (mol kg$^{-1}$) was prepared by dissolving $NaClO_4$ in water. This neutral electrolyte served as the base solution for subsequent preparations. Alkaline electrolytes were obtained by adding 0.1, 0.2, 0.4 and 0.8 mL of 1 M NaOH solution to 30 mL of the neutral electrolyte. For subsequent full cell testing, the optimized alkaline electrolyte prepared with the addition of 0.4 mL of 1 M NaOH was used unless mentioned otherwise.

### Coating preparation

The coating was prepared as follows: 0.1 g Nafion-Na was dissolved in mixed solution of 0.45 g N, N-Dimethylformamide and 0.45 g isopropanol at 60 °C. 0.025 g Ni/C and magnetically stirred for 0.5 h, and ultra-sounded for 0.5 h. Procedures were replicated three times to give an even mixture. 10 µL cm$^{-2}$ solution was sprayed on the surface of the cathode discs. Following removal of the solvent at room temperature (RT) (*ca.* 25 °C) in $N_2$-filled glove box under vacuum over 24 h, the electrode discs were coated homogenously.

### Electrochemical measurement

LSV was measured in a three-electrode cell at a scanning speed of 1 mV s$^{-1}$. In the three-electrode cell, glass carbon was used as the working electrode, Ti as the counter electrode, and an Ag/AgCl electrode as the reference electrode, respectively. The three-electrode cell for testing cathodes was assembled with NMF composite (20 mg cm$^{-2}$) as the working electrode, activated carbon as the counter electrode, and Ag/AgCl as the reference electrode. The three-electrode cell for testing anodes was assembled with the NTP composite (19 mg cm$^{-2}$) as the working electrode, activated carbon as the counter electrode, and Ag/AgCl as the reference electrode. All battery and electrochemical energy storage tests are conducted in an RT environment at *ca.* 25 °C, unless mentioned otherwise.

### Characterizations

In situ Attenuated Total Reflectance Infrared (ATR-IR) spectroscopy was determined using a Thermo-Fisher Nicolet iS20 equipped with a liquid nitrogen-cooled HgCdTe (MCT) detector, using a VeeMax III ATR accessory (Pike Technologies). A germanium prism (60°, Pike Technologies) was fitted within a PIKE electrochemical three-electrode cell comprising an Ag/AgCl reference electrode (Pine Research) and a Pt-wire counter electrode. To minimize potential influences from the electrolyte, nanocarbon particles, and polymer support, a background spectrum was acquired without applying voltage. The tests were determined utilizing a three-electrode cell with Pt serving as the counter electrode and Ag/AgCl as the reference. Differential electrochemical mass spectrometry (DEMS) was used to monitor volatile gases of $H_2$ and $O_2$ produced during battery operation at RT (Hidden HPR40). XRD patterns were determined using a Bruker-AXS Micro-diffractometer (D8 ADVANCE) with Cu-Kα1 radiation (λ = 1.5405 Å). HAADF-STEM, EDS mapping and line-scan spectra were used to confirm existence of Ni in cycled NMF particles (FEI Titan Themis 80-200). A cross-section of coated NMF cathode was determined via field emission (FE) focused ion-beam (FIB, Helios NanoLab 600), and data collected by SEM and EDS with a field emission scanning electron microscope (FEI Quanta 450). ICP-MS were collected using Agilent 8900 (the sample had been diluted 1000 times). Soft XAS was tested in the Australian Synchrotron. *Operando* synchrotron X-ray powder diffraction was conducted in the Australian Synchrotron and, the battery was tested by Neware battery test system (CT-3008-5V1mA-164, Shenzhen, China). Home-made 2032-coin cells were used for data collection. Both sides of the cell cases were punched with a central, 5 mm diameter hole, and sealed with Kapton film as the beam entrance. XRD data from the synchrotron were refined via the Rietveld method using GSAS II software.

### Density functional theory (DFT) computations

DFT computations were determined via Vienna Ab Initio Simulation Package (VASP.5.4.4)[22,23]. Generalized gradient approximation (GGA) with Perdew-Burke-Ernzerhof (PBE) function was used for describing the exchange-correlation potential[24,25]. Ni(111) and Ni(200) slabs were modelled using 3 × 3 unit cells with four layers, whilst the two topmost layers were allowed to fully relax until the convergence criterion of 10$^{-5}$ eV for energy and 0.02 eV/Å for final forces on atoms, whilst other layers were fixed. Energy cut-off was set as 600 eV. DFT-D3 correction method was used for describing the van der Waals interaction. A (3 × 3 × 1) Monkhorst-Pack k-point grid mesh was used. An implicit solvent model was used to simulate the solvent environment using Polarizable Continuum Model (PCM) provided by VASPSOL[26,27], where $\varepsilon_r$ = 80 for the water system.

For the adsorption energy ($\Delta G_{ads}$) for $*H_2O$, the reference state was set based on the liquid phase computed from, namely: $G_{H_2O(l)} = G_{H_2O(g)} + RT \times \ln(p/p_0)$, and $G_{H_2O(g)}$ determined via DFT computation. The adsorption energy for other absorbents, including OH and H, was computed from: $G_{OH^-} = G_{H_2O(l)} - G_{H^+}$ and; $G_{H^+} = 1/2 G_{H_2} - k_B T \ln 10 \times pH$, with pH = 12 used throughout.

$\Delta G_{ads}$ for a charge-neutral surface ($\Delta G_{ads}^{cnm}$) was computed from:

$$\Delta G_{ads}^{cnm} = G_1 - G_0 - G_{H^+} + |e|U \qquad (2)$$

where $G_1$ and $G_0$ denote, respectively, free energy for Ni surface with and without species adsorption and U is applied voltage *vs.* SHE model. The superscript cnm represents charge-neutral conditions. For the charge effect from applied voltage, an alternative constant-potential DFT (CP-DFT) can be reliably used. With this, the Fermi energy for the catalyst prior to ($E_{Fermi}^{Q_0}$) and following adsorption ($E_{Fermi}^{Q_1}$) using electron energy ($\mu_e$) is as follows[28]:

$$E_{Fermi}^{Q_0} = E_{Fermi}^{Q_1} = \mu_e \qquad (3)$$

where $\mu_e$ is determined from applied voltage, therefore is U dependent.

To satisfy Eq. (3) a manipulation of surface charge prior to and following adsorption ($Q_0$ and $Q_1$) is needed to maintain a specific Fermi energy. For this, $\Delta G_{ads}$ under CP-DFT ($\Delta G_{ads}^{cp}$) can be rewritten as[29]:

$$\Delta G_{ads}^{cp} = G_1^{Q_1} - G_0^{Q_0} - G_{H^+} - (Q_0 - Q_1) \times \mu_e \qquad (4)$$

where $G_1^{Q_1}$ and $G_0^{Q_0}$ are free energy obtained with CP-DFT. The catalyst is embedded in the electrolyte with a Poisson-Boltzmann model implemented via VASPSOL that balances the added charge by counter ions in solution and therefore obviates using a charged unit cell.

For DFT computation of the Ni substitution energy, to account for strong correlation effects in the 3d orbitals of Mn, Ni and Fe, a Hubbard U correction was included with the values of 4.0, 5.5 and 4.0 eV, respectively. The kinetic energy cut-off for plane wave expansion was set at 800 eV in all computations.

The initial NMF structure was constructed using 2 × 2 × 2 supercells with Fe and Mn atoms alternatively occupying metal sites. To simulate the dissolution of Mn atom, one Mn atom from the initial NMF structure was intentionally removed before geometry optimization. To simulate the doping with Ni atom, a Mn atom was substituted by a Ni atom in the initial NMF structure, followed by geometry optimization. For geometry optimization, the Brillouin zone was sampled with a (3 × 3 × 3) grid of k-points mesh with a Gamma-centered Monkhorst-Pack scheme. The structure was relaxed until energy and force converged below 10$^{-6}$ eV and 0.05 eV/Å, respectively.

## Data availability

Data that support findings from this study are available from the corresponding author on reasonable request. The source data underlying

Figs. 1–5 are provided as a Source Data file. Source data are provided in this paper. Source data are provided with this paper.

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

## Acknowledgements

The authors gratefully acknowledge financial support from the Australian Research Council (DP220102596, IL230100039, IH200100035, LP210301397 and DE230100471). DFT calculations were undertaken with the assistance of resources and services from the National Computational Infrastructure (NCI) and Phoenix High Performance Computing, supported by the Australian Government and, The University of Adelaide. This research was undertaken on the XAS and Soft X-ray beamlines at the Australian Synchrotron, part of ANSTO. The authors acknowledge Dr. Ashley Slattery for assistance with TEM testing, and TEM measurements were undertaken at Adelaide Microscopy, Centre for Advanced Microscopy and Microanalysis. The authors thank Tingting Liu from the Qingdao Institute of Bioenergy and Bioprocess Technology for assistance with DEMS testing.

## Author contributions

S.-Z.Q. conceived and supervised the project. H.W. and J.H. designed experiment, conducted characterizations and electrochemical measurement. Y.J., J.L. and X.X. conducted partial characterizations of materials. H.W., S.-Z.Q., K.D., C.W. and J.H. analysed data and wrote the manuscript. Y.J. conducted DFT computations. H.W. and J.H. contributed equally to this work. All authors agreed on the manuscript and approved submission.

## Competing interests

H.W. and S.-Z.Q. have filed a PCT provisional patent covering materials and sodium aqueous battery applications described in this manuscript. The remaining authors declare no competing interests.
