## [Peer Review File · Nature Communications]

Alkaline-based aqueous sodium-ion batteries for large-scale energy storageEditorial note: This manuscript has been previously reviewed at another journal that is not operating a transparent peer review scheme. This document only contains reviewer comments and rebuttal letters for versions considered at *Nature Communications*.

REVIEWER COMMENTS

Reviewer #1 (Remarks to the Author):

The manuscript suggests that NTP is more stable in an alkaline electrolyte, whereas NMF behaves in the opposite way. To ensure the cyclic stability of the cathode, the author proposes constructing a Ni/C coating. The manuscript demonstrates the excellent cyclic stability of NTP/NMF battery. However, the explanation of the mechanism is deemed unreliable. Therefore, the reviewer thinks this manuscript can be accepted for publication after major revisions due to the following reasons:

- 1) The authors suggest that a cathode surface layer rich in hydrated protons is formed. The question arises as to whether hydrated protons can enter NMF instead of sodium ions, and if so, whether this is the root cause of the superior rate performance?
- 2) The authors claim that Ni/C coating can prevent OH⁻ from diffusing from the bulk phase to the electrode surface because of the presence of Nafion. Is it possible to design experiments to prove that this Coating material can achieve the barrier effect on OH⁻?
- 3) Is it necessary to use PB material for the positive electrode, or would the NiOOH/Ni(OH)₂ reaction suffice in achieving the same excellent electrochemical performance?
- 4) In Figure 3a, why does the O-H stretching signal come from hydrated protons rather than OH⁻? Supplementary Figure 15a data suggest that it is challenging for hydrating protons to exist in a certain amount in the alkaline electrolyte, as there were no observable peak signal.
- 5) The poor reversibility of the high voltage platform is evident from the charge-discharge curve in Figure 3b. How to explain the inconsistency with Figure 4d?

Reviewer #2 (Remarks to the Author):

In the revised revision, the authors addressed some of the concerns raised by reviewers, but some issues have not been well addressed. For example, comments 1-1 and 1-3, 2-4, 2-5, 2-8, and 2-15. Additionally, the underlying mechanisms for the Ni-doping process and improved cycling stability are not well discussed. More revisions need to be considered.

1. It is hard to confirm whether the insertion of hydrated protons into NMF happened.
2. When evaluating the capacity of the electrodes, the mass of Ni/C coating or NiOOH/Ni(OH)₂ should be taken into account.
3. What's the energy density of the pouch cells?
4. In Fig. S9d-f, the reversibility of NMF electrodes in the half cells is poorer than that in the full cells. Why?
5. What's the driving force of Ni doping in the NMF, it is questionable why Ni²⁺ entered the NMF and located at Mn vacancies. What's the formation energy of NMF and Ni-doped NMF?
6. The proposed Fig. 4a is just a hypothesis, more characterizations should be provided.
7. The Na₃V₂(PO₄)₃@C is reported to show only one pair of redox peaks, why two cathodic peaks are observed in this work in Fig. S23?

RESPONSE TO REVIEWS

Response to Reviewer #1

Reviewer's general comments

The manuscript suggests that NTP is more stable in an alkaline electrolyte, whereas NMF behaves in the opposite way. To ensure the cyclic stability of the cathode, the author proposes constructing a Ni/C coating. The manuscript demonstrates the excellent cyclic stability of NTP/NMF battery. However, the explanation of the mechanism is deemed unreliable. Therefore, the reviewer thinks this manuscript can be accepted for publication after major revisions due to the following reasons:

Response

We thank Reviewer #1 for his/her positive comments, and recommendation for publication.

Comment 1-1

The authors suggest that a cathode surface layer rich in hydrated protons is formed. The question arises as to whether hydrated protons can enter NMF instead of sodium ions, and if so, whether this is the root cause of the superior rate performance?

Response

Following the Reviewer's suggestion, we conducted X-ray Photoelectron Spectroscopy (XPS) and Cyclic Voltammetry (CV) tests to establish the possibility of proton insertion. Given the significantly smaller size of protons compared with Na⁺, the co-intercalation of proton will significantly impact the local Mn environment, affecting the XPS patterns for Mn. However, as is seen in **Figure R1**, there is no significant difference in the Mn 2p peaks between the cathodes cycled with, and without, Ni/C coating. Importantly, this finding evidences that protons do not replace the position of sodium during cycling.

Considering the smaller size of protons, insertion of them would be expected to significantly alter battery reaction kinetics. To determine the electrochemical kinetics for the cathodes with and without Ni/C coating, we analyzed CV curves at differing scanning rates (**Figure R2**). It is widely accepted that the current consisting of diffusion and capacitive is given by:

$$i = av^b \quad (1)$$

where v is scan rate ($V s^{-1}$), i is peak current (A) and a and b are adjustable parameters. A ' b ' value = 1 indicates that the surface capacitance dominates charge storage, and a value 0.5 is indicative of mass diffusion-controlled charge storage^{1,2}. The b value remains constant after applying the Ni/C coating, evidencing that the reaction kinetics for the battery have not undergone significant change.

The number of protons induced by Ni coating at the interface is not sufficient to exhibit fully reversible proton insertion, because to establish reversible electrode cycling with a capacity of 118 mA h g⁻¹ at the cathode, a minimum proton concentration of 4.4×10⁻⁵ mol cm⁻² is needed. Based on computations however, the total protons triggered by Ni at the interface would be < 1.7×10⁻⁶ mol cm⁻². Therefore, the number of protons induced by Ni coating at the interface is not sufficient to exhibit reversible proton insertion.

The greater rate performance of batteries following Ni/C coated we attribute to the ‘large’ lattice parameters following Ni introduction. The Rietveld refinements for NMF with/without Ni/C coating after 1st cycle confirm that both electrodes exhibit cubic phases with Fm-3m space group and a = b = c (**Figure R3** and **Table R1**), however, the lattice parameters for Ni/C coated NMF (5.28161 Å) are greater than for uncoated NMF (5.26358 Å). This finding is attributed to the introduction of Ni in Ni/C coated NMF on cycling. An increased a (b, c) contributes to a boosted rate performance for the cathode.

In response to address fully this comment of Reviewer #1 we have in our Revised Manuscript (R-MS), p. 13, included the following explanatory text:

‘The lattice parameters for Ni/C coated NMF (5.28161 Å) are greater than for uncoated NMF (5.26358 Å). This finding is attributed to the introduction of Ni in NMF on cycling. An increase in a (b, c) contributes to a boosted rate performance for the cathode, which, significantly, is consistent with our findings (Supplementary Fig. 9a).’

Figure R1 XPS pattern for cycled NMF electrode.

Figure R2 Relationship between peak current and sweep rate in CV tests for with, or without, Ni/C layer in alkaline electrolyte. **a**, $\text{Fe}^{2+}/\text{Fe}^{3+}$ redox peak **b**, $\text{Mn}^{2+}/\text{Mn}^{3+}$ redox peak. b -value is computed from the slope of the plot.

Figure R3 Structural and Rietveld refinements for NMF cathode following 1st cycle in **a**, Alkaline electrolyte and **b**, Alkaline electrolyte with Ni/C coating.

Table R1 Detailed structural information on samples cycled for 1st cycle following Rietveld refinements.

Sample	$a=b=c$ (Å)	$\alpha=\beta=\gamma$	Volume (Å ³)	Size (μm)	Rwp (%)
NMF cathode following cycling in alkaline electrolyte	5.26358	90°	145.829	0.0454	5.36
Ni/C coated NMF cathode following cycling in alkaline electrolyte	5.28161	90°	147.333	0.2229	5.35

Comment 1-2

The authors claim that Ni/C coating can prevent OH from diffusing from the buck phase to the electrode surface because of the presence of Nafion. Is it possible to design experiments to prove that this Coating material can achieve the barrier effect on OH?

Response

We agree with Reviewer #1.

We designed experiments to establish the barrier effect of this coating material on the diffusion of OH^- . An H-cell configuration, illustrated in **Figure R4**, was used to establish the coating layer barrier effect(s). An alkaline electrolyte with a pH of 12.3 was introduced into the left-side of the H-cell, whilst a neutral electrolyte with a pH of 6.5 was added to the right-side, in which the uncoated cellulose, Nafion-Na coated cellulose, and Ni/C coated cellulose were used as separators, respectively. During the resting phase, OH^- ions in the left-side of the H cell migrate to the right-side resulting in an increase in pH in the right-side. To quantify this effect, we used a pH meter to determine the rate of pH increase in the right-side. As seen in **Figure R5**, in comparison to the uncoated cellulose, the Nafion-Na coated cellulose exhibits a suppression of OH^- ion migration, and the Ni/C boosts this effect.

Figure R4 Digital photo of the H-cell to test barrier effect of coating.

Figure R5 Increasing pH rate in OH^- penetration tests using H-cells.

In response therefore to directly address this comment of Reviewer #1 we have in our Revised Supporting Information (R-SI), p. 4, included the additional text:

*‘Experiments were conducted to establish the barrier effect of this coating material on OH⁻ migration. An H-cell configuration, illustrated in **Supplementary Fig. 16**, was used. An alkaline electrolyte with a pH of 12.3 was introduced into the left-side of the H-cell, whilst a neutral electrolyte with a pH of 6.5 was added to the right-side, in which the uncoated cellulose, Nafion-Na coated cellulose, and Ni/C coated cellulose were used as separators, respectively. During the resting phase, OH⁻ ions in the left-side of the H-cell migrate to the right-side, resulting in an increase in pH in the right-side. To quantify this effect, we used a pH meter to determine the rate of pH increase in the right-side. As seen in **Supplementary Fig. 17**, in comparison with the uncoated cellulose, the Nafion-Na coated cellulose exhibits a suppression of OH⁻ ion migration, and the Ni/C boosts this effect.’*

Comment 1-3

Is it necessary to use PB material for the positive electrode, or would the NiOOH/Ni(OH)₂ reaction suffice in achieving the same excellent electrochemical performance?

Response

We coated Ni/C into the surface of carbon paper and used this directly as the cathode. As can be seen in **Figure R6**, the capacity of this battery fades ‘quickly’ with Coulombic efficiency < 90 %.

Figure R6 Cycling performance for Ni/C used directly as a cathode.

We also tested Ni(OH)₂ cathode in the alkaline electrolyte. The Ni(OH)₂ electrode was prepared *via* mechanically mixing 80 wt.% Ni(OH)₂, 10 wt.% Super-P and 10 wt.% polytetrafluoroethylene (PTFE) binder dispersed in ethanol solvent (AR). The mixture was pressed on a Ti-mesh at a pressure of 6 MPa, and dried at 70 °C for 2 h in a vacuum-oven. As

can be seen in **Figure R7**, the cycling performance of Ni(OH)₂ in this alkaline electrolyte is ‘worse’ than that for Ni/C, a finding that can be attributed to the lower carbon concentration in the electrode. Therefore, irrespective of a Ni/C or Ni(OH)₂ cathode, it is not sufficient to exhibit an equivalent excellent electrochemical performance.

Figure R7 Cycling performance for Ni(OH)₂ used directly as a cathode.

Comment 1-4

In Figure 3a, why does the O-H stretching signal come from hydrated protons rather than OH⁻? Supplementary Figure 15a data suggest that it is challenging for hydrating protons to exist in a certain amount in the alkaline electrolyte, as there were no observable peak signal.

Response

In this work the O-H stretching signals were analysed based on a typical reference that especially studies the hydrated proton using IR³ e.g. Towards complete assignment of the infrared spectrum of the protonated water cluster H⁺(H₂O)₂₁, *Nat. Commun.* 12, 6141 (2021). This article reportedly used a combination of theoretical calculations and experiment to establish the peak assignment of hydrated protons in IR (**Figure R8**). Findings confirmed that these peaks at 1230, 1,798 and 2,032 cm⁻¹ are attributable to hydrated protons, rather than OH⁻. For our work, the same peaks are shown in IR, and therefore come from hydrated protons rather than OH⁻.

The intensity of peaks related to protons in Raman spectra are significantly weaker compared with those in IR spectra. However, minor peaks at higher voltage *ca.* 1.3 V (**Figure R9**) are still observed. Reasons for this are attributed to the difference in working mechanisms of the two types of *in-situ* experiment, i.e. Raman and IR. The accumulation of hydrating protons occurs under the coating layer. Based on the working mechanism of *in-situ* IR and *in-situ* Raman, the IR testing region is beneath the coating layer (**Figure R10**). The intensity of

peaks related to hydrated protons is therefore ‘relatively’ strong in IR spectra. Conversely, the Raman testing region is located directly on the surface of the coating layer (as seen in **Figure R11**), therefore exposed directly to the alkaline environment. As a result, the peaks related to hydrated protons are significantly weaker in the Raman spectra compared with that in the IR spectra.

In response therefore to address fully this comment of Reviewer #1 we have in our R-MS removed the Raman testing.

Figure R8 IR spectra for $H^+(H_2O)_{21}$ cluster.

Figure R9 *In-situ* Raman spectra for Ni/C in alkaline electrolyte (with D_2O as solvent).

Figure R10 Structure of *in-situ* IR cell and IR testing area.

Figure R11 Structure of *in-situ* Raman cell and Raman testing area.

Comment 1-5

The poor reversibility of the high voltage platform is evident from the charge-discharge curve in Figure 3b. How to explain the inconsistency with Figure 4d?

Response

As is seen in **Figure 3b** we used DEMS cells to run the battery. Based on its configuration (**Figure R12**) it cannot be sealed entirely and requires a continuous flow of argon to transport the generated gas into the Mass Spectrometer for analysis.

The data for **Figure 4d** in contrast were determined using well-sealed coin cells with a stable, internal pressure. It is expected therefore that the cycling performance in **Figure 3b** would be significantly less than that in coin cells because of these differences in sealing, and pressure control. However, **Figure 3b** presents monitoring of gas production during battery cycling.

It is concluded therefore that variation(s) in cycling stability at a high voltage do not materially affect the accuracy of the findings presented (in this figure).

Figure R12 Structure of DEMS cell.

Response to Reviewer #2

Reviewer's general comments

In the revised revision, the authors addressed some of the concerns raised by reviewers, but some issues have not been well addressed. For example, comments 1-1 and 1-3, 2-4, 2-5, 2-8, and 2-15. Additionally, the underlying mechanisms for the Ni-doping process and improved cycling stability are not well discussed. More revisions need to be considered.

Response

We thank Reviewer #2 for his/her considered comments, and agree.

A. In response the comments raised by Reviewer 1 & 2 in previous version are directly addressed as follows:

[Redacted]

B. In the following are the responses to the new comments raised by Reviewer #2.

Comment 2-1

It is hard to confirm whether the insertion of hydrated protons into NMF happened.

Response

Please *see* detailed response to Comment 1-1, p. 2.

Comment 2-2

When evaluating the capacity of the electrodes, the mass of Ni/C coating or NiOOH/Ni(OH)₂ should be taken into account.

Response

Please see detailed response to (Previous) Comment 1-3, and response to (Previous) Comment 1-3, p. 6.

Comment 2-3

What's the energy density of the pouch cells?

Response

We computed the energy density of the pouch cell.

The assembling of our pouch cells is at laboratory-level. This laboratory battery assembling prevents us from using thinner current collectors, lighter separators, lower carbon content and higher electrode compaction densities. As a result, the energy density remains relatively low.

The total mass of the pouch cell is *ca.* 6.7 g. The energy density therefore is *ca.* 10.89 Wh kg⁻¹ based on the total battery mass. However, because this value is determined in laboratory it does not represent potential of the system for larger, practical conditions.

The cell-level energy density of our aqueous sodium battery was also computed based on a virtual cell configuration with realistic parameters are tabulated in **Table R3**. The compacted density and porosity of the electrode, the amount of electrolyte, the size of the current collector, tab and package, were determined using empirical parameters reported in the literature, assuming a 10-layer pouch cell geometry. The energy density was computed *via* dividing the total energy by the total mass of the pouch cell. The predicted battery energy density is *ca.* 61 Wh kg⁻¹, which, importantly, is competitive amongst all aqueous batteries.

In response to address fully this comment of Reviewer #2 we have in in our R-MS, p. 6, included the following explanatory text:

*'The cell-level energy density of our ASIB based on a virtual cell configuration with realistic parameters as tabulated in **Supplementary Table 3** was computed. The compacted density and porosity of the electrode, the amount of electrolyte, size of the current collector, tab and package, were determined using empirical parameters from the literature assuming a 10-layer pouch cell geometry. The energy density was computed via dividing the total energy by the total mass of the pouch cell. The predicted battery energy density is *ca.* 61 Wh kg⁻¹.'*

Table R3 Data for computation of cell-level energy density.

Cell component	Parameter	Type/amount	Unit
Cathode	Active material (AM)	NMF	
	AM capacity	116	mAh g ⁻¹
	Coating mass	0.03	g cm ⁻²
	Loading	95.50 %	
	Binder	PTFE ¹	
	Conductive agent	Super P	
	Ni/C coating	1 %	
	Current collector	Ti foil	
	Thickness	6	μm
Anode	Active material (AM)	NTP	
	AM capacity	120	mAh g ⁻¹
	coating mass	0.03070845	g cm ⁻²
	Loading	96.50 %	
	Binder	PTFE	
	Conductive agent	Super P	
	Current collector	Ti foil	
	Thickness	6	μm
Electrolyte	Salt	NaClO ₄	
	Solvent	H ₂ O	
	Electrolyte/Capacity Ratio	2.5	g Ah
Separator	Celgard		
	Mass	1.09	mg cm ⁻²
	Length	10.6	cm
	Width	10.6	cm
Cell geometry	Size of CE ²	10*10	cm ²
	Size of AE ³	10.3*10.3	cm ²
	Size of tabs	1	cm ²
	Size of Separator	10.6*10.6	cm ²
	N/P Ratio	1.07	
	Size of Al-plastic film	12.6*25.2	cm ²
	Cathode Layers	10	
	Anode Layers	11	
	Separator Layers	22	
	Mass of Ni Tabs	0.2167	g cm ⁻²
Number of Ni Tabs	2	ea	
Cell	Output Voltage	1.46	V
	Capacity	6.64	Ah
	Total mass	159	g
	Energy density	61.0	Wh kg ⁻¹

Notes:

¹ PTFE: polytetrafluoroethylene² CE: cathode electrode³ AE: anode electrode

Comment 2-4

In Fig. S9d-f, the reversibility of NMF electrodes in the half cells is poorer than that in the full cells. Why?

Response

The difference between half-cell systems and full-cell systems is in the amount of electrolyte used.

In a half-cell we used three-electrode cells, as is illustrated in **Figure R27**. In this configuration a minimum of 3 mL of electrolyte is required for battery testing. However, full cells were tested within coin cells that contain just 0.1 mL of electrolyte. The reversibility of NMF cathode is closely related to electrolyte volume, particularly with metal dissolution.

Therefore, the half-cell that uses a significantly greater volume of electrolyte, exhibits lower reversibility compared with the full battery system.

Figure R27 Digital photograph of three electrodes cell used for half-cell tests.

Comment 2-5

What's the driving force of Ni doping in the NMF, it is questionable why Ni^{2+} entered the NMF and located at Mn vacancies. What's the formation energy of NMF and Ni-doped NMF?

Response

We used DFT calculations (**Figure R28**) to establish the driving force behind Ni substitution.

In Mn-based PBA cathode, the dissolution of Mn leads to creation of Mn vacancies, consequently diminishing stability of the cathode^{5,8,9}. We therefore initiated DFT calculations *via* removing a Mn atom from the original structure and compared its formation energy with that of a vacancy already occupied by a Ni atom. The DFT calculations gave a ΔE value of -8.06 eV, evidencing that Ni substitution is a spontaneous reaction.

Figure R28 DFT calculations for Ni substitution energy.

In response to directly address this comment we have in our R-MS:

1) p. 11, included additional explanatory text, namely;

*‘DFT calculations for Ni substitution were conducted. As can be seen in **Supplementary Fig. 20**, this predicted a ΔE value of -8.06 eV, evidencing that Ni substitution is spontaneous.’*

2) p. 19;

‘For DFT calculation of the Ni substitution energy. To account for strong correlation effects in the 3d orbitals of Mn, Ni, and Fe, a Hubbard U correction was included with the values of 4.0 eV, 5.5 eV and 4.0 eV, respectively. The kinetic energy cut-off for plane wave expansion was set at 800 eV in all calculations in this work.’

The initial NMF structure was constructed by constructing $2 \times 2 \times 2$ supercells with Fe and Mn atoms alternatively occupy metal sites. To simulate the dissolution of Mn atom, we intentionally removed one Mn atom from the initial NMF structure and then performed geometry optimization. To simulate the doping with Ni atom, a Mn atom was substituted by a Ni atom in the initial NMF structure, followed by geometry optimization. For geometry optimization, the Brillouin zone was sampled with a $(3 \times 3 \times 3)$ grid of k -points mesh with a Gamma-centred Monkhorst-Pack scheme. The structure was relaxed until energy and force converge below 10^{-6} eV and 0.05 eV/Å, respectively.’

Comment 2-6

The proposed Fig. 4a is just a hypothesis, more characterizations should be provided.

Response

We agree with this comment of Reviewer #2 that more characterizations should be provided.

We therefore conducted multiple characterizations to confirm our hypothesis.

1. Mn dissolution

During charging the surface of the Mn-based PBA cathode undergoes Mn dissolution^{5,8,9}, resulting in the creation of Mn vacancies. This phenomenon was established using Scanning Transmission Electron Microscopy-Energy-Dispersive Spectroscopy (STEM-EDS).

Analyses confirmed that Mn dissolution occurs in both neutral and alkaline electrolyte (**Figure R29**).

Figure R29 STEM-EDS mapping for cycled NMF electrode in neutral electrolyte (top), alkaline electrolyte (middle), and Ni/C coating (bottom).

2. Ni substitution

We conducted DFT calculations for Ni substitution. As can be seen in **Figure R30**, this predicted a ΔE value of -8.06 eV, evidencing that Ni substitution is a spontaneous reaction.

Several selected characterizations were made to establish successful doping of Ni including, *operando* Raman spectra, STEM-EDS mapping and line scan. In Raman spectra (**Figure R31a**), prior to cycling, two peaks at 2089 and 2124 cm^{-1} were apparent corresponding to $\text{Fe}^{2+}\text{-CN-Mn}^{2+}$ and $\text{Fe}^{2+}\text{-CN-Mn}^{3+}$ vibrations¹⁰, respectively. After charging to 1.89 V, both peaks ‘disappeared’, evidencing transformation of Fe^{2+} to Fe^{3+} and Mn^{2+} to Mn^{3+} . Importantly, after charging to 2.2 V, a new ‘weak’ peak appeared at 2195 cm^{-1} corresponding to $\text{Fe}^{3+}\text{-CN-Ni}^{2+}$ ⁷.

This finding confirms the introduction of Ni atom in NMF particle following transformation of Mn^{2+} to Mn^{3+} . After discharging to 0.5 V, peaks for $\text{Fe}^{2+}\text{-CN-Mn}^{2+}$ and $\text{Fe}^{2+}\text{-CN-Mn}^{3+}$ shift to 2092 and 2128 cm^{-1} and a new peak appeared at 2164 cm^{-1} that is assigned to $\text{Fe}^{2+}\text{-CN-Ni}^{2+}$.

Introduction of Ni is also evidenced *via* the STEM-EDS mapping (Figure R31b). There is a new peak belonging to Ni in the spectra. Additionally, the EDS line scan spectra for a single NMF particle confirms that Ni atoms are introduced into the edge of particles to suppress dissolution of inner Mn atoms (Figure R31c).

Figure R30 DFT calculations for Ni substitution energy.

Figure R31 a, *Operando* Raman spectra for Ni/C coated NMF cathode cycled in alkaline electrolyte. **b**, STEM-EDS spectra taken from NMF electrodes following cycling in neutral, alkaline electrolyte, and alkaline electrolyte with Ni/C coating. **c**, STEM line scan for cycled NMF cathodes with Ni/C coating. Inset shows Ni mapping for NMF.

In response therefore to directly address this comment we have in our R-MS, p. 10, reorganized relevant text to now read as follows, namely:

*'In addition to increased OER, the improved alkalinity of electrolyte compromises the cycling stability of PBA-based cathode material (without Ni/C coating). Although the cathode dissolution in alkaline electrolyte can be suppressed by the H_3O^+ -rich local environment, this cannot be obviated through pH modification because of the significant Mn dissolution in neutral, or acid, electrolyte. Beyond inducing H_3O^+ -rich environment, the Ni/C coating result in substitution of Ni to fill Mn vacancies, thereby stabilizing the PBA cathode. The working principle for this is illustrated in **Fig. 4a**. During charging, the surface of the Mn-based PBA cathode exhibits Mn dissolution, leading to the generation of Mn vacancies¹⁷. In the unprotected system, the continuous dissolution of Mn ions leads to the structural collapse and adversely affects the cyclic stability of the battery (**Figs. 4b-c**). However, in the Ni/C protected system the in-situ substituted Ni atoms balance 'tiny' structural disturbances caused by Mn dissolution as was evidenced by the stable discharge plateau of NMF/NTP battery following the coating^{10, 18, 19} (**Fig. 4d**). The oxidation of Ni during discharge produces Ni^{2+} ions. These Ni^{2+} ions will 'gradually' enter the crystalline framework to fill the Mn vacancies by forming Ni-N bonds on cycling. DFT calculations for Ni substitution were carried out. As can be seen in **Supplementary Fig. 20**, these gave a ΔE value of -8.06 eV, evidencing that Ni substitution is spontaneous.*

*Ni substitution was confirmed via operando Raman spectra for Ni/C coated NMF cathodes cycled in alkaline electrolyte (**Fig. 4e**). Prior to cycling, two peaks at 2089 and 2124 cm^{-1} were apparent, corresponding to $Fe^{2+}-CN-Mn^{2+}$ and $Fe^{2+}-CN-Mn^{3+}$ vibrations, respectively²⁰. After charging to 1.89 V both peaks 'disappear', evidencing the transformation of Fe^{2+} to Fe^{3+} and Mn^{2+} to Mn^{3+} . Importantly, after charging to 2.2 V, a new 'weak' peak appeared at 2195 cm^{-1} corresponding to $Fe^{3+}-CN-Ni^{2+}$ ²¹. This finding confirmed that the introduction of Ni atom in NMF particle follows transformation of Mn^{2+} to Mn^{3+} . After discharging to 0.5 V, peaks of $Fe^{2+}-CN-Mn^{2+}$ and $Fe^{2+}-CN-Mn^{3+}$ shifted to 2092 and 2128 cm^{-1} and a 'new' peak appeared at 2164 cm^{-1} that is assigned to $Fe^{2+}-CN-Ni^{2+}$ ²¹, confirming introduction of Ni.*

*The introduction of Ni was confirmed via the STEM-EDS mapping (**Fig. 4f**). There is new peak belonging to Ni element in the spectrum. Additionally, the EDS line scan spectra for a single NMF particle confirms that Ni atoms are introduced into the edge of particles to suppress dissolution of inner Mn atoms (**Fig. 4g**). The STEM-energy-dispersive spectroscopy (STEM-EDS) mappings for NMF cathode with Ni/C coating following 1st, 5th*

and 20th cycles (**Supplementary Table 6** and **Supplementary Fig. 21**) evidence that the content of Ni in NMF particles is stable after the first cycle, confirming that the introduction of Ni into NMF cathode reaches an equilibrium in the first cycle to give long-term stability to the battery.

To establish the boosted cathode structure stability after Ni/C protection, other characterizations were conducted. Digital photographs of the PBA electrodes following cycling showed that the unprotected electrodes exhibit significant metal ion dissolution in both neutral and alkaline condition (**Supplementary Fig. 22**). In comparison this is significantly mitigated when the Ni/C protective coating is applied. TEM analysis evidenced that the structure of the PBA cathode undergoes meaningful damage following cycling in alkaline condition (**Supplementary Fig. 23**). STEM-EDS confirmed that the electrode exhibits Mn dissolution in the neutral media, whilst the concurrent dissolution of both Fe and Mn occurs in alkaline electrolyte (**Supplementary Fig. 24**). However, following application of the protective coating this practical problem of electrode dissolution is significantly suppressed resulting from in-situ Ni substitution.’

Comment 2-7

The Na₃V₂(PO₄)₃@C is reported to show only one pair of redox peaks, why two cathodic peaks are observed in this work in Fig. S23?

Response

We conducted a critical literature investigation and an experiment to establish this ‘abnormal’ phenomenon.

Actually, observation of additional small cathodic peaks in this voltage range (~1.7 V vs Na/Na⁺) is not unique to our MS, but rather is common when testing NVP electrodes.

To establish this observation, we conducted tests with NVP in a conventional organic electrolyte, 1 M NaClO₄ in DMC:EC at a 1:1 volume ratio, with the addition of 5 % FEC, as shown in **Figure R32**. This experiment exhibited the presence of an additional cathodic peak. Significant is that similar observations are reported by Wang *et al* and Cui *et al*, as illustrated in **Figure R33** and **Figure R34**, respectively. This (collective) evidence establishes that the appearance of these ‘extra’ cathodic peaks in this voltage range is established phenomenon, and not specific to our MS. We observe also that the small peaks in the aqueous-based electrolyte are ‘more pronounced’ than those in the organic electrolyte.

We tentatively hypothesize that this can be attributed to water molecules. (We intend a more detailed analysis in future work).

Figure R32 Cyclic voltammety (CV) curves for NVP@C nanofibers measured in organic electrolyte.

Figure R33 Cyclic voltammety (CV) curves for NVP@C nanofibers measured in conventional organic electrolyte (1 M NaClO₄ in EC/PC/5 vol% FEC) and 19 m NaClO₄-NaOTF at a scan of 0.1 mV s⁻¹ 11.

Figure R34 CV curves for NVP@C at 1 mV s⁻¹ 12.

References

- 1 Augustyn, V. *et al.* High-rate electrochemical energy storage through Li⁺ intercalation pseudocapacitance. *Nature Materials* 12, 518-522 (2013).
- 2 Liu, B.-T. *et al.* Extraordinary pseudocapacitive energy storage triggered by phase transformation in hierarchical vanadium oxides. *Nature Communications* 9, 1375 (2018).
- 3 Liu, J. *et al.* Towards complete assignment of the infrared spectrum of the protonated water cluster H⁺(H₂O)₂₁. *Nature Communications* 12, 6141 (2021).
- 4 Mimura, H., Lehto, J. & Harjula, R. Chemical and thermal stability of potassium nickel hexacyanoferrate (II). *Journal of Nuclear Science and Technology* 34, 582-587 (1997).
- 5 Lamprecht, X., Speck, F., Marzak, P., Cherevko, S. & Bandarenka, A. S. Electrolyte effects on the stabilization of Prussian blue analogue electrodes in aqueous sodium-ion batteries. *ACS Applied Materials & Interfaces* 14, 3515-3525 (2022).
- 6 Nakamoto, K., Sakamoto, R., Ito, M., Kitajou, A. & Okada, S. Effect of concentrated electrolyte on aqueous sodium-ion battery with sodium manganese hexacyanoferrate cathode. *Electrochemistry* 85, 179-185 (2017).
- 7 You, Y., Wu, X.-L., Yin, Y.-X. & Guo, Y.-G. A zero-strain insertion cathode material of nickel ferricyanide for sodium-ion batteries. *Journal of Materials Chemistry A* 1, 14061-14065 (2013).
- 8 Ge, J., Fan, L., Rao, A. M., Zhou, J. & Lu, B. Surface-substituted Prussian blue analogue cathode for sustainable potassium-ion batteries. *Nature Sustainability* 5, 225-234 (2022).
- 9 Jiang, L. *et al.* Building aqueous K-ion batteries for energy storage. *Nature Energy* 4, 495-503 (2019).
- 10 Asahara, A. *et al.* Growth dynamics of photoinduced phase domain in cyano-complex studied by boundary sensitive Raman spectroscopy. *Acta Physica Polonica A* 121, 375-384 (2012).
- 11 Jin, T. *et al.* High-energy aqueous sodium-ion batteries. *Angewandte Chemie International Edition* 60, 11943-11948 (2021).
- 12 Liu, T. *et al.* Water-locked eutectic electrolyte enables long-cycling aqueous sodium-ion batteries. *ACS Applied Materials & Interfaces* 14, 33041-33051 (2022).

END OF RESPONSE TO REVIEWS

REVIEWERS' COMMENTS

Reviewer #1 (Remarks to the Author):

The manuscript suggests that NTP is more stable in an alkaline electrolyte, whereas NMF behaves in the opposite way. To ensure the cyclic stability of the cathode, the author proposes constructing a Ni/C coating. The manuscript demonstrates the excellent cyclic stability of NTP/NMF battery. H₃O⁺-rich local environment is proved to exist by In-situ surface-enhanced and other characterization methods. In-situ electrode Ni modification is fully demonstrated. Other comments made earlier were well responded. So, this manuscript will be suggested for publication in nature communication.

Reviewer #2 (Remarks to the Author):

The authors have addressed most of the concerns, the quality of the manuscript has been improved.